# Bluetongue Virus Nonstructural Protein 3 Orchestrates Virus Maturation and Drives Non-Lytic Egress via Two Polybasic Motifs

**DOI:** 10.3390/v11121107

**Published:** 2019-11-29

**Authors:** Thomas Labadie, Sophie Jegouic, Polly Roy

**Affiliations:** Department of Infection Biology, London School of Hygiene and Tropical Medicine, London WC1E 7HT, UK; thomas.labadie@lshtm.ac.uk (T.L.); sophie.jegouic@lshtm.ac.uk (S.J.)

**Keywords:** Bluetongue virus, virus budding, virus assembly, non-structural protein, polybasic motif

## Abstract

Bluetongue virus (BTV) is an arthropod-borne virus that infects domestic and wild ruminants. The virion is a non-enveloped double-layered particle with an outer capsid that encloses a core containing the segmented double-stranded RNA genome. Although BTV is canonically released by cell lysis, it also exits non-lytically. In infected cells, the BTV nonstructural glycoprotein 3 (NS3) is found to be associated with host membranes and traffics from the endoplasmic reticulum through the Golgi apparatus to the plasma membrane. This suggests a role for NS3 in BTV particle maturation and non-lytic egress. However, the mechanism by which NS3 coordinates these events has not yet been elucidated. Here, we identified two polybasic motifs (PMB1/PMB2), consistent with the membrane binding. Using site-directed mutagenesis, confocal and electron microscopy, and flow cytometry, we demonstrated that PBM1 and PBM2 mutant viruses retained NS3 either in the Golgi apparatus or in the endoplasmic reticulum, suggesting a distinct role for each motif. Mutation of PBM2 motif decreased NS3 export to the cell surface and virus production. However, both mutant viruses produced predominantly inner core particles that remained close to their site of assembly. Together, our data demonstrates that correct trafficking of the NS3 protein is required for virus maturation and release.

## 1. Introduction

Bluetongue virus (BTV) is an arthropod-borne virus transmitted by biting midges of the *Culicoides* genus. Since the 1950s, BTV has spread globally from Africa, through southern Europe, Asia and North America, and is currently endemic worldwide, causing periodic outbreaks with high morbidity, often with significant mortality and is consequently associated with substantial economic losses in the agricultural industry. Among the virus hosts, sheep and white-tailed deer exhibit the most severe clinical disease, characterized by ecchymosis, cardiac lesions, and hemorrhages [1,2,3], while BTV infection in other ruminants predominantly induces subclinical infection [4,5].

BTV is a member of the *Orbivirus* genus within the *Reoviridae* family. Like other members of the family, the virion is a non-enveloped icosahedral particle. BTV consists of seven structural proteins and a genome of 10 segments of double-stranded RNA (dsRNA). The seven structural proteins (VP1–VP7) are organized into two distinct capsids, with an outer capsid formed of VP2 and VP5 which surrounds an inner capsid (or core) formed of VP3 and VP7 [6]. The remaining three structural proteins, VP1, VP4, and VP6 comprise the polymerase complex, which is assembled into the inner core together with the 10 genomic dsRNA segments [7]. In addition to these structural proteins, BTV also synthesizes five non-structural proteins, NS1, NS2, NS3/NS3A, NS4, and NS5 that facilitate virus replication [8]. In particular, the glycoprotein NS3 has been shown to be involved in virus egress, although the mechanism of this role was not defined [9].

One characteristic of BTV infection in mammalian cells is that although the majority of mature virions are released by cell lysis in late infection, some particles are released from infected cells by local extrusion and budding from the plasma membrane. The NS3 protein is the only glycoprotein of BTV with two transmembrane domains. It has been predicted that this protein forms homo-oligomers in infected cells [10] and may have viroporin activity [11]. During the infection, the NS3 protein is expressed in the endoplasmic reticulum and then traffics through the Golgi apparatus before reaching the plasma membrane [12,13]. In virus-infected cells, several molecular interactions have also been reported between NS3 and cellular factors involved in exocytosis [12,14,15] and between NS3 and the outer capsid proteins [16,17]. Moreover, NS3 closely associates with newly synthesized progeny viruses, and subsequent studies have shown that NS3 is involved in the virus budding process [13,15,18,19]. Recently, it has been shown that NS3 also plays a role in supporting virus replication, by activating the MAPK/ERK signaling pathway [20].

However, the mechanisms by which NS3 reaches the plasma membrane to facilitate virus budding have not been fully elucidated. In this study, we identified two polybasic motifs (PBM1 and PBM2) in the NS3 protein that are conserved throughout the *Orbivirus* genus and may act as membrane export signals. Direct mutagenesis of the PBMs in the replicating genome revealed that PBM1 and PBM2 have two distinct signaling roles and are involved in trafficking through the endoplasmic reticulum and the Golgi apparatus. Additionally, mutation in PBM2 decreased the level of NS3 surface expression. Interestingly, infected cells with PBM1 mutant viruses (NS3_PBM1_) produced mainly core particles that remained close to their site of assembly, whereas PBM2 mutant viruses (NS3_PBM2_) produced core particles that were distributed through the cytoplasm. In consequence, BTV release was delayed significantly in cells infected by PBM mutant viruses, and only core particles were released. Together, our data demonstrates that PBM are responsible for correct trafficking of the NS3 protein, allowing non-lytic release of mature particles.

## 2. Materials and Methods

### 2.1. Cell Lines and Virus Stocks

BSR cells (a derivative of the Baby Hamster Kidney cells BHK21 [21] were maintained in Dulbecco’s Modified Eagle’s Medium (DMEM) (Sigma-Aldrich, St. Louis, MI, USA), supplemented with 5% foetal calf serum (FCS) and antibiotics (100 units/mL penicillin, 100 mg/mL streptomycin, GIBCO, Life Technologies). The BSR/NS3 stable cell line, constitutively expressing the BTV NS3 protein, was grown in DMEM–5% FCS supplemented with 7.5 µg/mL of puromycin (Sigma-Aldrich). All cells were incubated at 37 °C in humidified 5% CO2 incubator. Virus stocks were obtained by infecting BSR cells at a multiplicity of 10^−1^ plaque-forming units per cell (pfu/cell) for 1 h under gentle agitation at room temperature. The inoculum was then removed and cells were maintained in DMEM with 2% FCS for 48 h. The harvested supernatants were clarified (4000 g × 10 min) and stored at 4°C. Stocks of the BTV-1 NS3_PBM1_ and BTV-1 NS3_PBM2_ viruses were obtained by infecting BSR-NS3 cells for 48 h with the same protocol, in order to obtain a higher viral titer. These mutant viruses were then always passaged on BSR cells before subsequent analysis.

### 2.2. Generation of Mutated Viruses and Cloning

Site-directed mutagenesis was performed to substitute each basic amino acid of the identified PBM1 and PBM2 (Figure 1B) toward alanine in the BTV-1 S10 sequence. Briefly, two complementary primers were used for mutating the concerned residues in pUC19-BTV1-T7-S10 or pcDNA-BTV1-S10 template by PCR with a KOD DNA polymerase (Merck Millipore, Burlington, MA, USA). The mutated versions of NS3 were sequenced (Eurofins Genomics, Luxembourg) to confirm the presence of the designed mutations. The mutant BTV and the WT BTV (BTV-1, European lineage) were rescued by reverse genetics, following the method described previously [22] and based on the transfection of T7 BTV-1 capped (S1-S9) and uncapped transcripts (S10, S10_PBM1_ and S10_PBM2_). All viruses had the same number of passages during the experiments, and segment 10 was sequenced (Sanger sequencing) to confirm the absence or presence of mutations.

### 2.3. Virus Titrations and Viral Replication Kinetic

Viral infectivity was estimated using plaque assay or a tissue culture infectious dose of 50 per mL (TCID_50_/mL) titration, for which viral suspensions were diluted from 10^−1^ to 10^−12^ in D-MEM medium in 96-well plates and titers were calculated according to the Spearman-Karber method. BSR-S10 cells were used for titration of the mutant NS3 virus. Intracellular virus titers were estimated using a TCID_50_ method on harvested cells after disruption by two freeze-thaw cycles.

### 2.4. Fluorescence Microscopy and Colocalization Analysis

BSR cells were seeded on glass cover slips in 6-well plates for 24 h before inoculation with BTV, at a multiplicity of 1 pfu/cell. After 24 h of infection, cells were washed twice with DMEM and fixed with a PBS–paraformaldehyde 2% (w/v) solution for 10 min at room temperature, followed by a step of membrane permeabilization with a PBS buffer containing 0.2% Triton X-100 for 15 min. A 30 min blocking treatment of fixed cells with a PBS-Bovine serum albumin (1% w/v) solution was complete prior to the overnight incubation at 4 °C of primaries anti-58K antibody (ab27043, Abcam, Cambridge, UK), anti-KDEL antibody (ab12223, Abcam), and antibodies against viral proteins NS3 (made in the Pr. Roy’s laboratory) diluted (1:1000) in PBS buffer (1% BSA). Coverslips were subsequently incubated at room temperature for 2 h with respective specific secondary antibodies coupled with Alexa Fluor 488, Alexa Fluor 546 and Alexa Fluor 633 (A11034, A11031, A11074, A21071, Thermo Fisher Scientific, Waltham, MA, USA) diluted at a concentration of 1:2000. Glass coverslips were then mounted on a glass slide with fluoromount aqueous mounting medium (F4680, Sigma-Aldrich). Confocal laser scanning of fluorescence was performed using a LSM880, inverted microscopes (Carl Zeiss Ltd., Oberkochen, Germany). Colocalization was analyzed by using the Pearson correlation coefficient, calculated for 30 selected regions of interest (ROI) from BSR cells infected with mutant or WT viruses, using the Icy colocalization studio plugin (https://icy.bioimageanalysis.org).

### 2.5. Protein Expression Levels by Flow-Cytometry

Cell surface protein expression of infected cells was measured by immunofluorescence using flow cytometry. BSR cells were seeded in 6-well plates for 24 h before the inoculation with BTV, at a multiplicity of 1 pfu/cell. After 24 h of infection, cell monolayers were washed twice with PBS, re-suspended using trypsin-EDTA (Thermo Fisher Scientific) and fixed with paraformaldehyde 2% (w/v) for 10 min. For each replicate, BSR cells were then split in two separate tubes. Cells permeabilization were performed with a solution of PBS-Triton (0.1% v/v) in only one of the two tubes, allowing subsequently to quantify total protein content in the permeabilized cells and cell surface protein only in non-permeabilized cells. The cells were then incubated with a rabbit anti NS3 and a mouse anti beta-actin (A5441, Sigma-Aldrich) antibodies overnight at 4°C, followed by an incubation of 2 h at room temperature in the presence of a species-specific Alexa Fluor 488 conjugated and Alexa Fluor 350 conjugated secondary antibodies (A11034 and A11045, Thermo Fisher Scientific). After centrifugation and washings, cells were re-suspended in PBS and analyzed with a BD LSR II flow cytometer (BD Biosciences, San Jose, CA, USA). For each replicate, 25,000 cells were analyzed using the same parameters. For the analysis, we used a gating strategy to discriminate doublet cells by plotting FSC-A vs. FSC-H, and selected NS3 positive cells using mock-infected cells stained the same way to determine the threshold of background fluorescence.

### 2.6. Analysis of Virus Sedimentation by Ultracentrifugation

Discontinuous gradients of iodixanol (Opti-Prep; Axis-Shield, United Kingdom) were prepared, using different concentrations in 10 mM Tris-HCl buffer (pH 8)–NaCl (0.85% w/v). Viral supernatants were clarified (2000× *g* for 10 min) and then layered onto an iodixanol gradient. The 14 mL tubes (344060, Beckman Coulter) were centrifuged at 150,000× *g* for 2 h at 4 °C using a Beckman SW 40 rotor, and 0.5 mL fractions were harvested manually from the top and analyzed for the presence of an infectious virus using a TCID_50_ titration method.

### 2.7. Western Blot

After ultracentrifugation, viral particles within the fractions of interest were lysed in Laemmli buffer at 95 °C for 5 min. Proteins were then separated on a 10% SDS gel and blotted on a PVDF membrane. The membranes were then cut horizontally and the primary anti VP2, VP5, VP7, and NS3/NS3a antibodies (made in the Pr. Roy’s laboratory) were incubated separately, followed by an incubation with respective Horseradish peroxidase-coupled secondary antibody (anti-guinea pig IgG ab97155 and anti-rabbit IgG ab97051, Abcam). Luminescence was then detected using SuperSignal West Pico PLUS Chemiluminescent Substrate (Thermo Fisher Scientific). The ImageJ software was used to perform linear contrast enhancement and protein semi quantification [23].

### 2.8. Transmission Electron Microscopy

BSR cells were infected at a MOI of 1 pfu/cell and processed for sectioning at 24 hpi. Briefly, after three washes with DMEM, monolayers were fixed in a solution of 2.5% paraformaldehyde,2.5% glutaraldehyde and 0.1% sodium cacodylate (pH 7.4), and post-fixed in 1% osmium tetroxide—0.1% sodium cacodylate. Cells were dehydrated in increasing concentrations of ethanol and embedded in epoxy resin (TAAB Laboratories Equipment Ltd., Aldermaston, United Kingdom). Ultrathin sections were stained with Reynolds lead citrate. For electron microscopy, infected cells were fixed with 4% paraformaldehyde followed by 2% formaldehyde, 0.1% glutaraldehyde, and 0.2% HEPES. Ultrathin slices were cut and mounted onto nickel grids. For virus particles analysis, purified WT NS3 virus was collected from fraction #3 and NS3_PBM1_ virus from fraction #4 after ultracentrifugation, adsorbed onto a carbon-coated copper grid, and stained with a 1% solution of uranyl acetate. Images were visualized with a transmission electron microscope (JEM-1010, JEOL Akishima, Tokyo, Japan).

### 2.9. Specific Infectivity

The specific infectivity is defined as the ratio between the number of infectious viral particles and the total amount of viral RNA. The number of infectious viral particles was determined by plaque assay, and an extraction of the viral RNA into the supernatant was performed to quantify the genome copy number. For this we used the GeneJET Viral DNA/RNA Purification Kit (K0821, Thermo fisher Scientific) followed by a reverse transcription using a universal degenerated oligo specific to all 5′ non-coding sequences of BTV segments (BTV/Uni1; 5′ GTTAAAWHDB 3′) and the GoScript Reverse Transcription System (Promega, Madison, WI, USA). The cDNA obtained for segment 10 was then quantified by a real time quantitative PCR, using the 2x SYBR Green qPCR Master Mix (Bimake, Houston, TX, USA) and S10 specific primers (BTV1 S10/249F, 5′ CATTCGCATCGTACGCAGAA 3′; BTV1 S10/464R, 5′ GCTTAAACGCCACGCTCATA 3′). For absolute quantification of the cDNA copy numbers, a standard curve was produced using a 10-fold serial dilution of a pUC 19 S10 plasmid of known concentration as a template for amplification.

### 2.10. Multiple Sequences Alignment of NS3 from Orbivirus Genus

NS3 sequences from members of the *Orbivirus* genus were obtained on GenBank, and the accession numbers are listed below. Amino acid sequences alignment was performed using Unipro UGENE, using a MAFFT program.

### 2.11. Statistical Analyzes and Software

Numerical data were analyzed with Prism software version 6.07 (Graph Pad Software, San Diego, CA, USA). A Grubb’s test was performed to identify and remove outliers in the data from the membrane protection assay. For all sets of data, variances were equal across samples (Bartlett’s test). According to the number of samples, two-tailed t-tests or ANOVA tests were performed for statistical comparison. Sanger sequencing analysis were performed using Unipro UGENE [24]. Microscopy imaging data were processed using LSM software Zen Blue edition version 2.3 (Carl Zeiss Ltd.) or Icy [25]. Flow cytometry data were analyzed with FlowJo (FlowJo LLC, Ashland, OR, USA) software.

## 3. Results

### 3.1. NS3 Proteins of Members of the Orbivirus Genus Share Two Conserved Polybasic Motifs

We have shown previously that while the cytoplasmic N-terminal region of the glycosylated NS3 protein interacts with cellular factors that are involved in exocytosis, the cytoplasmic C-terminal region interacts with viral outer capsid proteins [12,14,15,16,17]. Thus, NS3 is believed to act as a molecular bridge between viral proteins and cellular proteins in order to facilitate virus egress (Figure 1A). However, the domains of NS3 facilitating the export of this protein to the cell plasma membrane are unknown. By aligning multiple NS3 amino acid sequences from 20 members of the *Orbivirus* genus (Table 1), we identified two clusters of basic amino acids, composed of lysine, arginine or histidine, which are present in all orbiviruses upstream of the first transmembrane domain of the NS3 (Figure 1B). We designated these two motifs as polybasic motif 1 (PBM1) and polybasic motif 2 (PBM2), encompassing the amino acids at positions 92 to 97 and positions 114 to 121 respectively.

In order to understand the role of the conserved PBM1 and PBM2 motifs in NS3, we substituted each basic amino acid in PBM1 (NS3_PBM1_) and in PBM2 (NS3_PBM2_) with neutral, alanine residues. Each mutant virus was then generated using our BTV reverse genetics (RG) system [22], and the growth of these two mutant viruses was compared with the wild type (WT) virus in mammalian BSR cells, by titrating both intracellular particles and particles released in the supernatant (Figure 2). At 24 h post infection (hpi), WT BTV reached 6.2 log_10_(TCID_50_/mL) in the cytoplasm of infected cells, whereas the intracellular titers of the mutant viruses, NS3_PBM1_ and NS3_PBM2_, were 4.7 log_10_(TCID_50_/mL) and 4.5 log_10_(TCID_50_/mL), respectively, which was significantly lower than that of the WT virus. The number of WT virus particles released into the supernatant was equivalent to the titer of the cytoplasmic viruses at all times. However, significantly fewer viral particles of the mutant viruses were released in the supernatant compared with their respective viral titers in the cytoplasm. This suggests that substitution of basic amino acids in the PBMs of NS3 might have decreased virus fitness by preventing virus release and cell-to-cell transmission. Moreover, virus recovery was not possible when we introduced mutations at both sites simultaneously, emphasizing that both PBMs are essential for virus fitness.

### 3.2. PBM1 and PBM2 Are Required for the Cellular Trafficking of NS3 and for Viral Particle Budding

To examine if mutations at PBM1 and PBM2 have altered the trafficking of NS3, BSR cells infected with each of the viruses were analyzed by immunofluorescence and confocal microscopy. The NS3 localization was visualized by confocal imaging in relation to endoplasmic reticulum and Golgi apparatus using specific markers, KDEL and 58K, respectively (Figure 3A,B). We then analyzed the colocalization between NS3 and these markers using the Pearson correlation coefficient (Figure 3C,D). We found a high level of colocalization between WT NS3 and the ER marker KDEL (Pearson coefficient R = 0.40 ± 0.06), and no colocalization with 58K, the Golgi apparatus marker (Pearson coefficient R = 0.09 ± 0.04). In contrast, NS3_PBM1_ mutant was found to be colocalized with 58K (Pearson coefficient R = 0.31 ± 0.11), but not with KDEL (Pearson coefficient R = 0.16 ± 0.06). The NS3_PBM2_ mutant had a similar phenotype of WT NS3, presenting a higher colocalization with KDEL (Pearson coefficient R = 0.29 ± 0.08) than with 58K (Pearson coefficient R = 0.10 ± 0.07). These results suggest that synthesized NS3_PBM1_ protein is arrested in the Golgi apparatus and that PBM2 motif is not essential for NS3 trafficking between the ER and Golgi apparatus. To examine if mutations at PBM1 and PBM2 have altered the export of NS3 at the plasma membrane, we quantified the amount of NS3 on the cell surface of non-permeabilized BSR infected cells and the amount of total NS3 in permeabilized cells. NS3 surface expression for each virus was then compared by calculating the ratio between the surface and total amounts (Figure 3E). BSR cells exhibited significantly more fluorescence at the cell surface when infected with NS3_PBM1_ virus compared with cells infected with WT virus. In contrast, we could not detect surface NS3 in cells infected with NS3_PBM2_ virus. This suggests that the NS3_PBM1_ export to the plasma membrane is more efficient than WT NS3 export and that mutation in the PBM2 motif altered NS3 export to the plasma membrane. We also quantified the levels of beta-actin in permeabilized and non-permeabilized cells (Figure 3F), as a control to confirm that cytoplasmic proteins are not readily detectable at the plasma membrane of non-permeabilized cells, to validate our approach. Overall, these results suggest that the PBM1 and PBM2 are distinct signaling motifs for NS3 trafficking in infected cells. The NS3_PBM1_ was more concentrated in the Golgi apparatus compared with WT NS3 and NS3_PBM2_, and was exported more efficiently to the plasma membrane, suggesting that this motif acts as an ER retention signal. In contrast, NS3_PBM2_ had similar localization as WT NS3, but was not exported to the plasma membrane, suggesting that PBM2 acts as a membrane export signal from the Golgi apparatus.

Subsequently, we analyzed cells infected with WT BTV, NS3_PBM1_ or NS3_PBM2_ by transmission electron microscopy (TEM). At 24 hpi, BSR cells infected with WT BTV showed extensive accumulation of particles close to the viral inclusion bodies (VIBs), the virus core assembly sites in the cytoplasm (Figure 4). Moreover, significant budding events at the plasma membrane were exhibited. In contrast, there was no virus budding observed at the plasma membrane of BSR cells infected with either of the two mutant viruses, although particles were visible in the cytoplasm and within or around VIBs. The size of these particles appeared to be smaller than virions, suggesting that these particles are likely to be immature cores. These data demonstrate that polybasic motifs in the NS3 protein of BTV are required for BTV particle release through budding.

### 3.3. NS3 Trafficking Defect is Associated with Release of Non-Mature Core Particles

The NS3 protein is known to interact with the outer capsid protein VP2 [16] and VP5 [17] to mediates the virus release. Therefore, mutations disrupting the trafficking of NS3 are expected to perturb the release of mature virus particles. To assess this hypothesis, we harvested WT, NS3_PBM1_ or NS3_PBM2_ BTV particles at 48 hpi from BSR infected cells and analyzed their sedimentation rate by ultracentrifugation on a 5% to 35 % discontinue gradient of iodixanol (Figure 5A). Nine fractions were collected for viral titration using a TCID_50_ method. We found the highest amount of WT viral particles in fraction #3 (15% iodixanol), with 5.7 log_10_(TCID_50_/mL), while the highest amount of NS3_PBM1_ and NS3_PBM2_ viruses were found in fraction #4 (25 % iodixanol), with 4.4 log_10_(TCID_50_/mL) and 3.6 log_10_(TCID_50_/mL) respectively. These results indicate that both NS3_PBM1_ and NS3_PBM2_ viruses predominantly generate particles with a higher density than the WT virus. In order to determine their molecular compositions, we performed a Western blot analysis on fractions #3 of WT BTV and on fractions #4 of BTV NS3_PBM1_ and BTV NS3_PBM2_, as these fractions were more concentrated in infectious virions (Figure 5B). Subsequently, viral proteins in these purified fractions were quantified and the amount of the outer capsid proteins was normalized by the amount of virus particles (based on the VP7 core protein). The results for WT BTV showed distinct bands for three major capsid proteins, the two outer capsid proteins, VP2 and VP5, and the core surface protein VP7 in addition to NS3 proteins (Figure 5C). However, while the core protein VP7 was detected in a significantly higher quantity in the mutant viruses compared with the WT virus, both outer capsid proteins were barely detectable. Thus, these results suggest that the mutant particles could be mainly core particles, rather than double-capsid particles. These findings were further confirmed by imaging the WT and NS3_PBM1_ BTV particles by transmission electron microscopy (Figure 5D) and measuring their mean diameter (Figure 5E). We observed that NS3_PBM1_ virus particles had a smoother surface and a significantly lower diameter than WT BTV particles, with an average diameter of 56.6 nm and 65.6 nm, respectively. Altogether, our data demonstrate that NS3 trafficking defect is associated with the release of non-mature core particles.

## 4. Discussion

Previous work has shown that BTV uses both lysis and budding modes for its egress from infected cells. In this study, we demonstrate how trafficking of the BTV glycoprotein NS3 within the cell may orchestrate virus release. We identified conserved PBMs in the NS3 protein of BTV and more widely across the *Orbivirus* genus. It has been reported that PBMs act as plasma membrane export signals for some enveloped viruses, such as Murine Leukemia virus [28] and also for a non-enveloped virus [29], but has not yet been reported for an arthropod-borne virus. Interestingly, PBMs are also well known for protein signaling in the endoplasmic reticulum and Golgi apparatus [30,31,32,33]. In this study, we described a similar function as a cellular trafficking signal for the PBMs in the NS3 protein of BTV.

Recent insights into the function of the BTV NS3 protein challenge the canonical view of BTV as a non-enveloped virus that would egress from cells through a lytic mechanism. During the replication, the NS3 protein may act as a molecular bridge between the newly assembled BTV particles and cellular factors hijacked to support virus release, for example via two late domain motifs, PPRY and PSAP, which are present in the N-terminus of NS3. The PPRY motif binds proteins of the E3 ubiquitin ligases NEDD4 family and is responsible for the aggregation of newly synthesized BTV particles in multi-vesicular bodies derived from late endosomes [12]. The PSAP motif binds the TSG101 protein, a member of the ESCRT protein family, and is essential for BTV release through budding [15]. Indeed when the PSAP motif is deleted, BTV particles remain tethered at the plasma membrane, highlighting the importance of this motif in the final step of BTV non-lytic release [13]. Further, it has been demonstrated that NS3 also interacts with the Annexin A2 protein, which then binds the phosphatidylinositol 4,5-bisphosphate (PI(4,5)P2) present in cell membranes [14], and that NS3 is present in cholesterol-enriched membranes [17], as well as in the envelope surrounding budding particles in BTV-infected cells [19].

When we replaced basic amino acids in the PBMs by neutral alanine, we observed that NS3 trafficking in the cytoplasm was perturbed and that NS3_PBM2_ export to the plasma membrane was inefficient. Consequently, BTV bearing these mutations in the NS3 exhibited a delay in virus particle production in infected cells, and it was not possible to identify assembled particles in the cytoplasm. The effect of these mutations was even more apparent on virus particle release, correlating with the absence of budding events at the plasma membrane. These results lead us to conclude that the trafficking of NS3 in infected cell is required both for virus maturation and for egress. In cells infected with NS3_PBM1_, we observed a high concentration of NS3 in the Golgi apparatus, compared to the WT NS3. In contrast, NS3_PBM2_ was more present in the endoplasmic reticulum but exhibited significantly less surface expression in comparison to the WT NS3, indicating PBMs have two distinct roles in NS3 cellular trafficking. PBM-mediated protein trafficking is an effective process to export proteins from the trans-Golgi network to specific cell compartments, such as the apical pole of polarized cells [31]. The role of PBM is also dependent on their position in the protein. In the case of the reptilian reovirus p14 protein, PBM motif acts as a Golgi export signal when proximal to the transmembrane domain, and becomes an endoplasmic retention signal when distal to the transmembrane domain [34]. In our study, both PBMs are membrane proximal, but in agreement with the reovirus p14 protein, the NS3_PBM2_ motif mediates NS3 export from the Golgi apparatus to the plasma membrane, and NS3_PBM1_ motif is an ER retention signal. Several mechanisms have been described for PBM-mediated signaling to the plasma membrane, such as a phospholipid-dependent recruitment of the PBM [33,35] or specific recruitment in the lipid raft at the plasma membrane [36]. However, the direct interaction between PBM and specific lipids of the plasma membrane is not sufficient, as described for the PBM in the Septin 9 protein that allows enrichment in phosphoinositides by specific binding, but the physical association between the membrane and the protein could be mediated by amphipathic helices adjacent to the PBMs [32]. Similarly, a putative coil-coiled motif has been reported between PBM1 and PBM2 of NS3 and would require further characterization, thus increasing the need for an atomic structure of NS3.

In our study, we observed that the NS3 trafficking defect was associated with the release of BTV particles containing fewer outer capsid proteins, and having a smaller diameter with a higher density compared with WT BTV particles. We also detected more VP7, the major core protein, by Western blot in NS3_PBM1_ and NS3_PBM2_ viruses, compared with the WT virus, and found that the WT virus fraction contained more particles that were infectious, compared to mutant viruses. These two results suggest that more viral particles, mainly core particles, were present in the purified fraction of NS3_PBM1_ and NS3_PBM2_ virus compared with the WT virus, but were less infectious. This is consistent with previous studies, which demonstrated that BTV core particles have a lower infectivity in mammalian cells [37] and that only core particles are observed by electron microscopy when cells are infected with a virus that lacks NS3 [27]. It is also noteworthy that in an early study, two different populations of WT BTV particles with the same properties that we describe here had been observed although no further studies were undertaken regarding their assembly mechanism or egress [38]. It is likely that the smaller and denser BTV particles described by these authors were released through an NS3-independent mechanism, similar to our mutant viruses. We assume that such a mechanism of release occurs during cell death, in the late stage of the infection.

In conclusion, our studies here describe how trafficking of NS3, the only membrane protein of the BTV, orchestrates a non-lytic pathway of egress for these non-enveloped viruses. Our results demonstrate that the polybasic domains of NS3 act as distinct trafficking signals, mediating endoplasmic reticulum retention of the protein, and Golgi apparatus export toward the plasma membrane. Moreover, NS3 trafficking during the infection is required for the release of complete mature particles. These novel findings demonstrate how arthropod-borne non-enveloped orbiviruses are released from the cell, identifying mechanisms that may have potential as new targets for intervention and control.

## Figures and Tables

**Figure 1 viruses-11-01107-f001:**
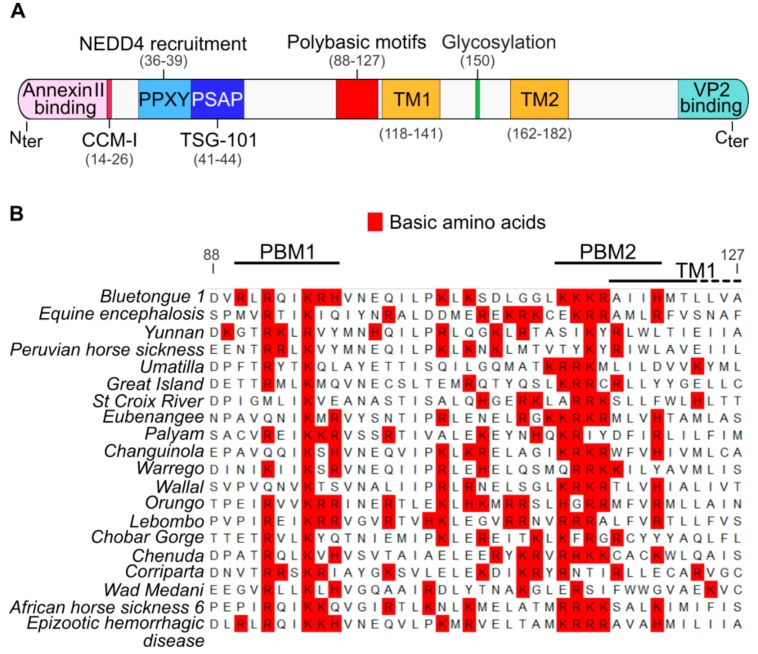
Identification of membrane proximal polybasic motifs that are conserved in the NS3 proteins of the *Orbivirus* genus. (**A**) Diagram presenting the main identified domains in the NS3 protein. From the N-terminal (N_ter_) to the C-terminal (C_ter_), the NS3 protein possess a Annexin II protein-binding motif, two late domains (PPXY/PSAP) binding respectively the E3 ubiquitin-protein ligase NEDD4 protein and the TSG-101 protein, a member of the ESCRT proteins family [12,13]. All these cellular interactors mediate virus release through budding. The NS3 protein also possess two transmembrane domains (TM1 and TM2) and a glycosylation site located between the TM [26]. The NS3 protein also binds the viral protein VP2 through a C-terminal motif [27]. The N-terminal coiled-coil motif I (CCM-I) is also presumed to be involved in NS3 protein oligomerization [10] (**B**) Multiple sequence alignment of all known *Orbivirus* NS3 amino acid sequences adjacent to the TM1 domain. Basic amino acids are highlighted in red. We named PBM1 and PBM2 the two most conserved polybasic domains. Amino acids numbering is based on the NS3 sequence of the BTV-1. Genbank accession numbers of each sequence are available in the material and methods section.

**Figure 2 viruses-11-01107-f002:**
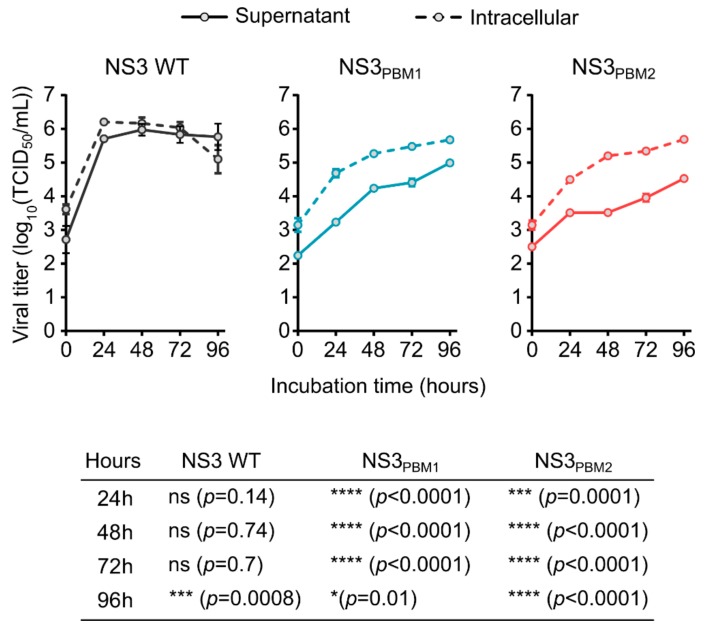
Substitution of the polybasic motifs of NS3 by neutral amino acids induce a decrease of viral fitness. Replication kinetic of intracellular BTV viral particles in infected BSR cells (dashed lines) and kinetic of viral particles release in the supernatant (solid line) for BTV WT virus (grey lines) or either BTV NS3_PBM1_ and BTV NS3_PBM2_ viruses (respectively blue and red lines). Each dot represents the mean viral titer, and vertical lines represent the standard deviation (*n* = 4). The bottom table summarizes the result from an ANOVA test comparing the titers of intracellular viruses with the supernatant viruses for BTV WT, NS3PBM1 and NS3BM2 viruses at different time points.

**Figure 3 viruses-11-01107-f003:**
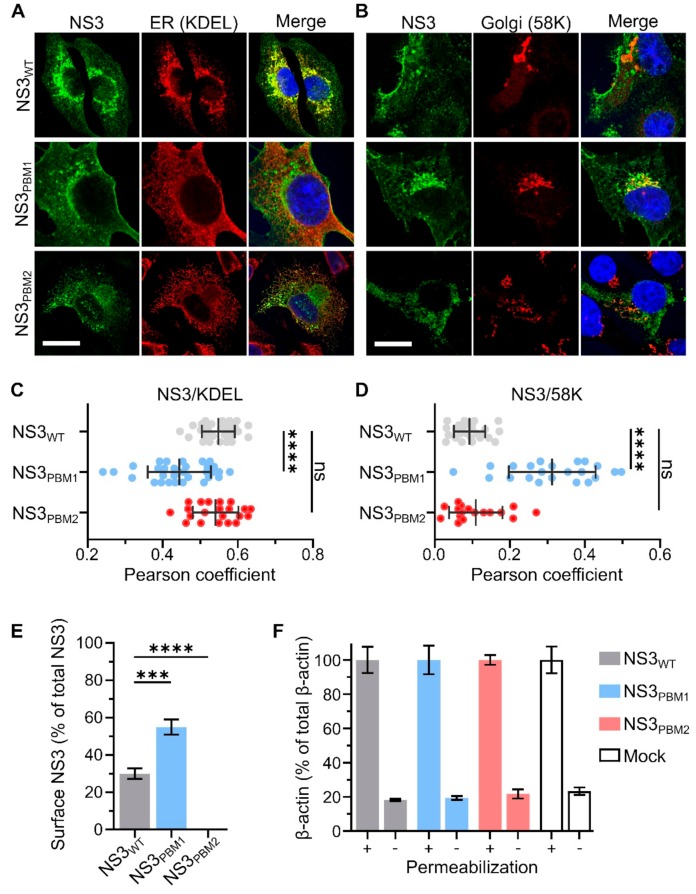
PBM1 and PBM2 are required for the cellular trafficking of NS3. (**A**,**B**) Confocal immunofluorescence microscopy (x63) of BSR cells 24 h after infection by either WT BTV (Top row), BTV NS3_PBM1_ (middle row) or BTV NS3_PBM2_ (bottom row) viruses. Fluorescence signal of the NS3 labelling is shown in green. The labelling of the endoplasmic reticulum (ER) marker (KDEL protein) is shown in red, as well as the Golgi apparatus marker (58K protein), and the nucleus is shown in blue. Scale bar represents 20 μm (white line, bottom left corner). The colocalization between NS3 and KDEL (**C**) and between NS3 and 58K (**D**) were quantified for each virus using the Pearson correlation coefficient. Horizontal lines represent the standard deviation. The mean Pearson correlation coefficients were compared using multiple Wilcoxon-Mann-Whitney Tests (ns *p*> 0.05; **** *p* < 0.0001; *n* = 30). (**E**) Median surface expression levels of the NS3 protein in infected BSR cells analyzed by flow cytometry detecting Alexa Fluor 488 signal 24 h after infection. The level of membrane-associated NS3 is expressed as the percentage of fluorescence level detected in non-permeabilized cells compared with the total fluorescence level detected in permeabilized cells. The means surface expression level of the NS3 protein was then compared using a t-test (*** *p* < 0.0005; **** *p* < 0.0001; *n* = 3). Vertical lines represent the standard deviation. (**F**) Negative control, the same experiment was performed to quantify the cytoplasmic beta-actin protein in BSR cells infected with the mutant and WT viruses, or not infected (mock). The «+» and «-» indicate permeabilized and not permeabilized prior to incubation with primary antibodies.

**Figure 4 viruses-11-01107-f004:**
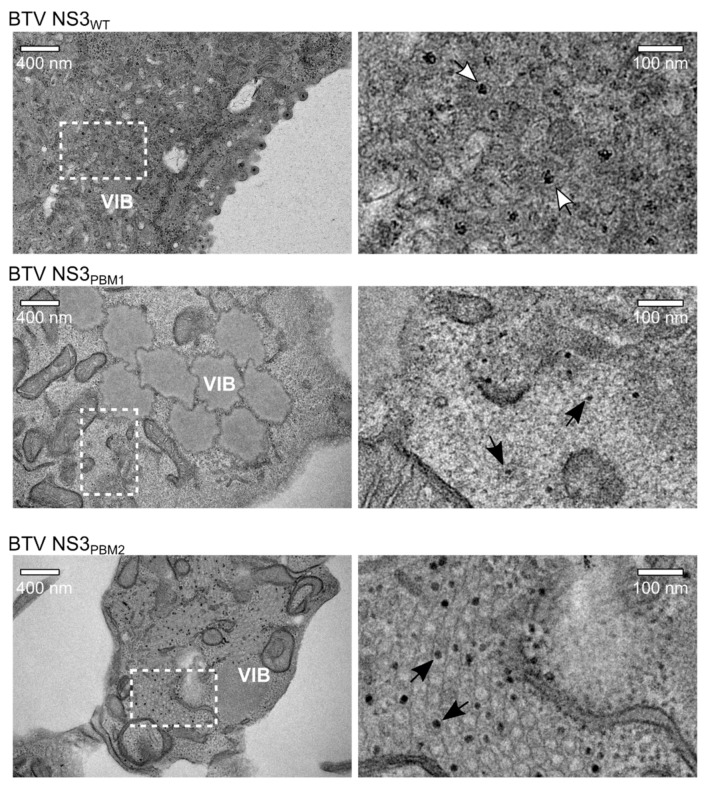
Transmission Electron Microscope analysis of infected BSR cells after sectioning. In BSR cells infected for 24h with WT BTV (top row), viral particles can be observed accumulating next to a VIB and budding at the plasma membrane, whereas we observed aggregated VIBs with only core particles inside and close to the viral inclusion bodies in BSR cells infected with BTV NS3_PBM1_ (middle row). In cells infected with BTV NS3_PBM2_ (bottom row), we observed an important accumulation of core particles within tubules. Images of the right panel are magnification views derived from white squares in images of the left panel. Dark solid arrows highlight the core particles; white open arrows highlight the mature virus particles.

**Figure 5 viruses-11-01107-f005:**
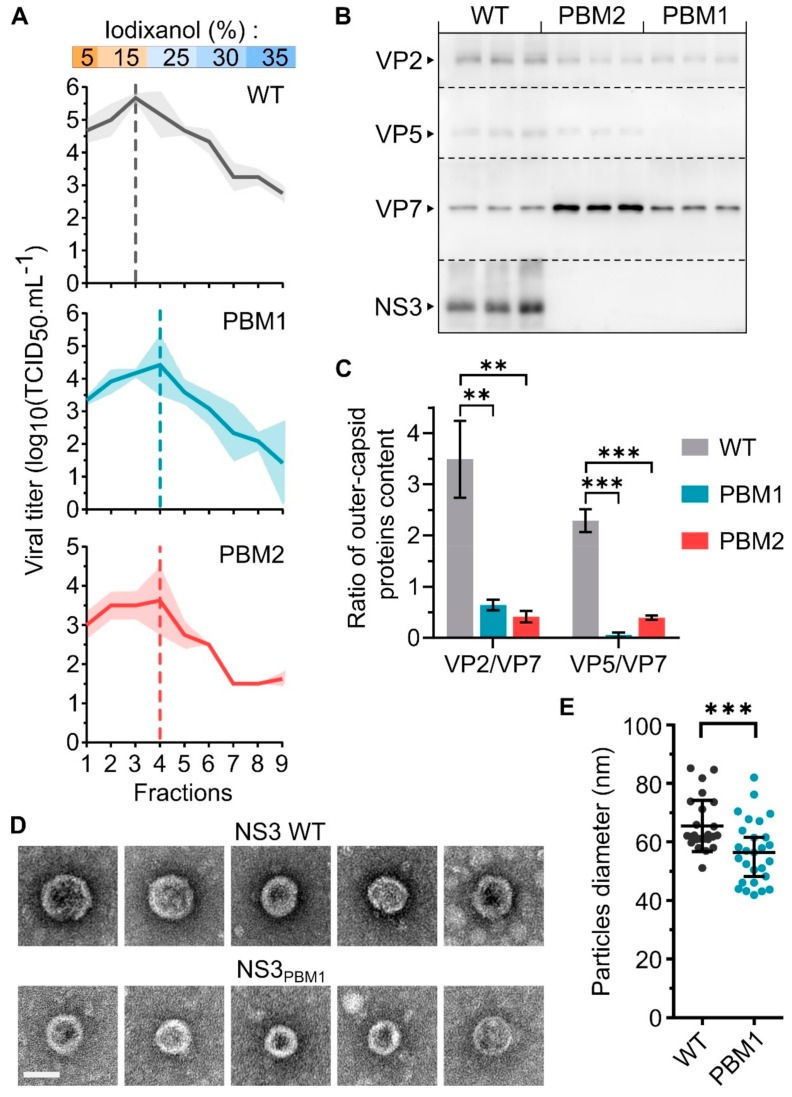
NS3 export to the membrane is required for the release of mature BTV particles. (**A**) Mean viral titer in each fraction harvested after ultracentrifugation of WT BTV (top), BTV NS3_PBM1_ (middle) and BTV NS3_PBM2_ (bottom) virus particles on a discontinuous iodixanol gradient. The density of the discontinuous iodixanol gradient used in this experiment is indicated on the top, and the shaded areas represent the standard deviation. Vertical dashed lines represent the fraction with the highest viral titer for each virus. (**B**) Analysis of the presence of BTV proteins VP2, VP5, VP7, and NS3/NS3a by Western blot analysis of the fraction #3 of WT BTV, and the fraction #4 of BTV NS3_PBM1_ and BTV NS3_PBM2_ after ultracentrifugation, each in three independent replicates. Dashed lines indicate the separation of the PVDF membrane after protein migration and transfer. (**C**) The level of each outer capsid protein was quantified and normalized with the level of core protein VP7, and the mean ratio were compared between each virus, using a two tailed t test (* *p* < 0.05; ** *p* < 0.005; *** *p* < 0.0005; *n* = 3). Vertical lines represent the standard deviation. (**D**) Transmission electron microscope imaging (×200.000) of purified BTV particles. Imaging of WT BTV particles are shown on top and BTV NS3_PBM1_ particles are shown below. Scale bar represents 50 nm (white line). (**E**) Mean diameter of BTV particles calculated using the ImageJ software and compared using a two tailed t-test (** *p* < 0.005, *n* = 30). Each dot represents a single measured diameter and vertical lines represent the standard deviation.

**Table 1 viruses-11-01107-t001:** Genbank accession numbers of all NS3 amino acid sequences from members of the *Orbivirus* genus used for multiple sequence alignment.

*Orbiviruses*	Genbank Accession Number
Bluetongue virus 1	ALI51183.1
Equine encephalosis virus	AEP95960.1
Yunnan virus	YP_443934.1
Peruvian horse sickness virus	YP_460047.1
Umatilla virus	YP_009047250.1
Great Island virus	YP_003896068.1
St Croix River virus	YP_052951.1
Eubenangee virus	AFH41518.1
Palyam virus	ALW83187.1
Changuinola virus	AIV43215.1
Warrego virus	AIT55722.1
Wallal virus	AIT55712.1
Orungo virus	AFX73397.1
Lebombo virus	AFX73386.1
Chobar Gorge virus	YP_009158911.1
Chenuda virus	YP_009158899.1
Corriparta virus	AGT51064.1
Wad Medani virus	YP_009158892.1
African horse sickness virus 6	AKP19850.1
Epizootic hemorrhagic disease virus	AAQ62564.1

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
