# Peer review of "Bluetongue Virus Nonstructural Protein 3 Orchestrates Virus Maturation and Drives Non-Lytic Egress via Two Polybasic Motifs"

_viruses, 2019, doi:10.3390/v11121107_

Round 1

Reviewer 1 Report

Line 89 write reverse genetics not "reverse genetic"

Legend figure 4, write VIB in full rather than using the abbreviation (that way the figure is comprehensible to everyone as a "stand alone")

Lone 396 say "fewer" rather than "less" (you are referring to a number).

Legend to Figure 2, line 234 , write "summarises" instead of "sum up". Sum up can mean "addition of" rather than a summary.

Figure 4. The regions shown in the magnified images must correspond to particular regions in the non-magnified images in the left hand column ( a small box could be superimposed on the 50K image to show from where the 200K image was extracted). Currently it is not possible to see that the magnified image is derived from part of the non-magnified image). Also it would be better to use the same magnification in all ages of the right hand column (for example in the second column it would be much better to use a 200K image, rather than 120K, so that direct comparison can be made with the images above and below). You need to re-do these images.

Author Response

We made the changes in the text asked by the reviewer.

According the reviewers’ suggestions, we also modified the figure 4. It now include magnifications derived from part of non-magnified images. The same magnifications were used for each panel (x50k on the left and x200K on the right), in order compare virus particles more easily. The selected images also allow to observe several WT viruses released at the plasma membrane, whereas no virus were observed at the plasma membrane in cells infected by NS3PBM1 and NS3PBM2 viruses.

Reviewer 2 Report

Overall this is a nicely designed study examining the role of the NS3 polybasic motifs in the egress and budding of BTV during non-lytic infection. The combination of techniques and quality of work are strengths and the paper is well written. I only have a couple minor concerns listed below.

1. Fig. 4 It appears that the 50K mag shots of the VIB’s are at different scales. In the NS3WT you can only see part of the cell whereas in the NS3PBM2 you can see the whole cell. Also there are no accompanying images of the plasma membrane for the two mutants.

2. Images of the western blots should be included either in the main text or as supplemental material.

Author Response

According the reviewers’ suggestions, we modified the figure 4. It now include magnifications derived from part of non-magnified images. The same magnifications were used for each panel (x50k on the left and x200K on the right), in order compare virus particles more easily. The selected images also allow to observe several WT viruses released at the plasma membrane, whereas no virus were observed at the plasma membrane in cells infected by NS3PBM1 and NS3PBM2 viruses.

In the revised version, we included a new western blot data in the Figure 5.

Round 2

Reviewer 1 Report

I suggest a small change in wording to the legend for Fig 4 . The authors have stated "viral particles can be observed accumulating next to a VIB (left) and budding at the plasma membrane (middle and right)." However, there is no middle image now, so the wording needs to be fixed. This error has happened because the images have been fixed as requested , but the this text has been carried over from the old version.

Otherwise OK.